# Synthesis of Co-Ni Alloy Particles with the Structure of a Solid Substitution Solution by Precipitation in a Supercritical Carbon Dioxide

**DOI:** 10.3390/nano12244366

**Published:** 2022-12-07

**Authors:** Nikolay Nesterov, Vera Pakharukova, Svetlana Cherepanova, Stanislav Yakushkin, Evgeniy Gerasimov, Dmitry Balaev, Sergei Semenov, Andrey Dubrovskii, Oleg Martyanov

**Affiliations:** 1Boreskov Institute of Catalysis, Siberian Branch, Russian Academy of Sciences, 630090 Novosibirsk, Russia; 2Kirensky Institute of Physics, Krasnoyarsk Scientific Center, Siberian Branch, Russian Academy of Sciences, 660036 Krasnoyarsk, Russia; 3Institute of Engineering Physics and Radioelectronics, Siberian Federal University, 660041 Krasnoyarsk, Russia

**Keywords:** supercritical fluids, Co-Ni alloy, solid substitution solution

## Abstract

Mixed Co-Ni bimetallic systems with the structure of a solid substitution solution have been synthesized using the supercritical antisolvent precipitation (SAS) method, which uses supercritical CO_2_ as an antisolvent. The systems obtained have been characterized in detail using X-ray diffraction (XRD), high-resolution transmission electron microscopy (HRTEM), Fourier-transform infrared (FTIR) spectroscopy, and magnetostatic measurements. It has been found that Co-enriched systems have a defective hexagonal close-packed (hcp) structure, which was described by a model which embedded cubic fragments of packaging into a hexagonal close-packed (hcp) structure. It has been shown that an increase in water content at the precipitation stage leads to a decrease in the size of cubic fragments and a more uniform distribution of them in Co-enriched systems. It has also been shown that mixed systems have the greatest coercivity in the line of samples. Ni-enriched bimetallic systems have a cubic close-packed (ccp) structure with modified crystal lattice parameters.

## 1. Introduction

Bimetallic alloy systems based on Co are of interest in various fields, such as the development of high-density recording devices [1], the use magnetic nanoparticles in medicine [2], and heterogeneous catalysis [3,4]. It is well known that metallic Co exhibits different structure polymorphs. It usually crystallizes as an 𝛼-Co phase with a hexagonal close-packed structure (hcp) and 𝛽-Co phase with a cubic close-packed (ccp) structure. The hexagonal 𝛼-Co phase is more stable at room temperature and with ambient pressure, while the cubic 𝛽-Co is thermodynamically stable above 450 °C [5,6]. Both polymorphs have closely packed atoms and differ in the stacking sequence (hexagonal -ABABAB- or cubic -ABCABC-). The real structure of metallic cobalt was found to depend on the particle size and synthesis route. Thus, it was shown that a decrease in particle size can lead to the formation of the cubic phase below 450 °C [7,8] or to the intergrowth of lamellar hcp and ccp fragments with the formation of nanostructured particles enriched by stacking faults [9,10].

The preparation of Co-containing catalysts based on mixed alloy systems with different compositions makes it possible to control the activity and selectivity of the process [11,12]. The ferromagnetic behavior of these systems opens an additional window of opportunity for their study by methods such as ferromagnetic resonance [13] and magnetodynamics research [14], which allows for information about the size and structural characteristics of the systems to be obtained through study of their magnetic properties. Bimetallic Co-Ni alloy systems have attracted researchers’ attention due to their catalytic activity [15] and their magnetic properties (coercive force, saturation magnetization, Curie temperature, etc.) [16].

Currently, various synthesis methods are applied to prepare Co-Ni alloy systems, such as mechanical alloying [17], sonochemical preparation [18], sol–gel synthesis [19], decomposition of acetylacetonates in the presence of surfactants [20], solution combustion synthesis [21], and the polyol method [22]. In the review conducted by the authors of [23], one can become acquainted with the methods of Co-Ni system synthesis in more detail.

Recently, alternative approaches for the synthesis of various functional materials have emerged, which are based on the use of supercritical fluid technologies [24]. For example, supercritical isopropanol was used for reduction of Co-containing catalysts [25], composites for CO_2_ capture from air were prepared through drying in supercritical ethanol [26], and gas separation membranes were obtained in supercritical CO_2_ [27]. Supercritical antisolvent (SAS) precipitation is one of the most promising methods for the synthesis of various functional materials used in catalytic transformations [28] and pharmaceutics [29,30]. Thus, in [31], Gd-doped ZnO photocatalysts for atrazine decomposition were synthesized using SAS. Single-domain powders of mixed europium and iron oxide with a garnet structure were obtained by the SAS method for the first time [32].

We have proposed a new approach to the preparation of metal and oxide systems by the SAS method, which is based on the co-precipitation of stable oxide sols [33,34]. This approach made it possible to synthesize dispersed Ni-containing catalysts [35,36,37] and to stabilize Au nanoparticles in alumina [38,39].

It was shown in [40] that the addition of water to a methanol solution makes it possible to avoid phase separation in Ni-Cu bimetallic systems obtained by SAS precipitation. In this work, we investigated the structure and magnetic properties of bimetallic Co-Ni systems with different ratios of metals when obtained by the SAS method. The effect of water as a co-solvent on the phase composition of Co-Ni bimetallic systems was studied.

## 2. Materials and Methods

### 2.1. Reagents

Nickel acetate tetrahydrate (Ni(OAc)_2_·4H_2_O, 99% extra, Acros Organics, Geel, Belgium), cobalt acetate tetrahydrate (Co(OAc)_2_·4H_2_O, ACS reagent, ≥98.0%, Sigma-Aldrich, Geel, Belgium), methanol (HPLC Gradient Grade, J.T. Barker, Deventer, The Netherlands), CO_2_ (99.8%, Promgazservis, Novosibirsk, Russia).

### 2.2. Co-Ni Sample Synthesis

The synthesis of Co-Ni samples was performed by supercritical antisolvent precipitation using a specially designed SAS-50 setup (Waters, Milford, MA, USA). The SAS apparatus was equipped with a precipitation chamber with a volume of 0.5 L. For more information about the apparatus scheme, see Appendix A. A methanol solution containing cobalt acetate and nickel acetate was injected into a stream of supercritical carbon dioxide. The solvent power of the carbon dioxide–methanol mixture became lower compared to pure methanol, leading to precipitation. After that, pure CO_2_ was passed through the obtained powder for 20 min to remove residual solvent. A detailed description of the synthesis technique is given in [33].

Experimental parameters: CO_2_ flow, 80 g/min; solution flow, 2 mL/min; temperature, 40 °C; nozzle, 0.004″ (0.10 mm); pressure, 150 bar. The total concentration of cobalt and nickel acetate was 15 mg/mL for all samples. These small concentrations allow us to be sure of the solution’s stability, since there are no traces of undissolved salts. The typical volume of the methanol solution was 80 mL. We obtained 7 species designated as CoXNiY_WZ, where X and Y represent the molar ratio of cobalt and nickel and Z describes the volume percentage of water as a co-solvent in methanol. Monometallic Co- and Ni-containing systems were synthesized, as well as three mixed systems with a ratio of Co/Ni = 2/1, 1/1, and 1/2. The samples after precipitation were calcined at 300 °C for 3 h with a ramp rate of 3 °C/min in static air to obtain the oxide phase. The reduction of samples was performed in a H_2_ atmosphere at 300 °C for 45 min with a ramp rate of 3 °C/min; the flow rate of H_2_ was 30 mL/min. We also synthesized a pure metallic cobalt sample by calcining cobalt acetate and reducing it in a H_2_ flow. The methods of calcination and reduction were the same as for the systems obtained by the SAS precipitation. This sample is designated as Co_AC.

### 2.3. XRD Characterization

The phase composition of the obtained Co-Ni samples was studied using a D8 Advance X-ray diffractometer (Bruker, Berlin, Germany) and Cu Kα radiation (λ = 1.5418 Å) with a step of 2θ = 0.05° and an accumulation time of 3 s at each point. The diffractometer was equipped with a linear LynxEye (1D) detector.

### 2.4. Simulation of X-ray Diffraction Patterns

To perform structural diagnostics of the imperfect metal, an approach based on the simulation of XRD profiles for a statistical model of one-dimensionally (1D) disordered crystal was applied. The program DEFECT was used [41]. The 1D-disordered crystal model was defined as a statistical sequence of a finite number of periodic 2D layers. X-ray scattering is known to be localized along the rods passing through hk nodes of a reciprocal 2D lattice. The scattering amplitudes were calculated along hk rods for each kind of layer and the intensity distribution along hk rods for a statistical sequence of layers was then calculated with the use of matrix formalism [42]. A Markov chain was used to generate a statistical sequence of layers. The finite size of layers and their shape was taken into account through the convolution of rod intensity with the squared module of a Fourier transform of the shape function.

Models of 1D-disordered crystals with the coherent inclusion of ccp fragments into the hcp matrix were generated with the use of one type of layer (AB) and two modes of stacking. The first stacking mode, (T_1_), generates an hcp structure with an AB-AB-AB sequence without any mutual shift of the layers. The second stacking mode, (T_2_), induces the appearance of cubic fragments with an AB-CA-BC sequence by shifting the AB layer on the vector (2/3, 1/3) with respect to the previous one. The models of 1D-disordered crystals with hcp–ccp-type intergrowths were defined by the following parameters: the fraction of layers stacked by the T_2_ mode (W_2_) and the conditional probability of cubic fragments appearing after the same previous one (P_22_). The conditional probabilities of the hexagonal type of stacking appearance after the same previous one (P_11_) and the changing type of close packing (P_12_, P_21_) can be easily calculated. To increase the thickness of ccp domains, it is necessary to increase the P_22_ parameter. Additional varied parameters were the crystallite sizes in the plane of the layers and the direction in which the layers were stacked.

The average sizes of the coherently scattering domains (CSDs), D_XRD_, were calculated by line broadening analysis according to the Scherrer equation.

### 2.5. HRTEM Characterization

The morphology and microstructure of the Co-Ni samples were studied via high-resolution transmission electron microscopy (HRTEM). The images were obtained using a Themis Z electron microscope (Thermo Fisher Scientific, Bleiswijk, The Netherlands) equipped with a Ceta 16 CCD sensor and a corrector of spherical aberrations, which provided a maximum lattice resolution of 0.07 nm at an accelerating voltage of 200 kV. The microscope was also equipped with an EDX Super-X spectrometer (Thermo Fisher Scientific, Bleiswijk, The Netherlands) with a semiconductor Si detector providing an energy resolution of 128 eV. The samples for the HRTEM study were deposited on a holey carbon film mounted on an aluminum grid by ultrasonic dispersal of the catalyst suspension in ethanol. The calculation of interplanar distances was carried out using fast Fourier transform (FFT) patterns with the help of Velox software (Version 2020, Thermo Fisher Scientific, Waltham, MA, USA) and Digital Micrograph (Version 2018, Gatan, Pleasanton, CA, USA).

### 2.6. FTIR Characterization

Fourier transform infrared (FTIR) spectroscopy was performed using a Bruker Vertex 70v spectrometer equipped with a diamond ATR accessory (Specac Ltd., Orpington, UK) and a DLaTGS detector. A total of 100 scans were taken for each sample and recorded from 4000 to 500 cm^−1^ at a resolution of 4 cm^−1^, and the spectra obtained are shown in ATR mode.

### 2.7. Magnetic Measurements

Magnetic measurements (the field dependencies of magnetization M(H)) were performed using vibrating sample magnetometers (VSMs) from the Quantum Design PPMS-6000 facility (operation temperature range 4.2–50 K, applied field up to 25 kOe), Lakeshore VSM 8604 (operation temperature range 77–400 K, applied field up to 15 kOe). For the data obtained in the 77–400 K range, sample demagnetization was performed at the beginning of every measurement.

## 3. Results and Discussion

### 3.1. Investigation of Structural Properties

In our previous work, we proposed an approach that uses water as a co-solvent for the synthesis of bimetallic systems, and it was shown that water addition into methanol makes it possible to obtain Ni-Cu metal systems with a substitutional solid solution structure without phase separation into individual metals [40]. In this work, we decided to use the proposed approach and synthesize a series of Co-Ni samples using 8 vol.% water as a co-solvent. The obtained samples were studied in the oxide state (after calcination) and the metallic state (after reduction).

Figure 1 shows the XRD spectra of the synthesized systems after calcination in air at 300 °C (I) and after reduction in a hydrogen flow at 300 °C (II). Table 1 shows the structural characteristics of Co-Ni-mixed systems. For Co-W8 and Ni_W8 samples after calcination, the XRD peaks of Co_3_O_4_ and NiO are respectively observed. The Co-enriched sample Co2Ni1_W8 is characterized by peaks of spinel-like phase on the base of Co_3_O_4_ containing Ni^2+^ ions. Unfortunately, it is not possible to determine Ni content in Co_3_O_4_ due to the close values of lattice parameters for Co_3_O_4_ (a = 8.083Å, PDF#42-1467) and NiCo_2_O_4_ (a = 8.11Å, PDF#20-0781). In contrast, the Ni-enriched sample Co1Ni2_W8 has a NiO-like structure that likely contains Co^3+^ ions since the lattice parameter is 4.129 Å, which is less than the standard NiO value of 4.177 Å [43] (the ionic radii of Co^2+^ and Co^3+^ cations in octahedral coordination are 0.65 and 0.55 Å, respectively). Of course, partial substitution of Ni^2+^ ions by Co^3+^ ones should lead to the appearance of vacancies. In the sample with the metal ratio of Ni/Co = 1/1, separation into NiO- and Co_3_O_4_-like phases takes place. It should be noted that the lattice parameter of NiO in the Co1Ni1_W8 sample is 4.129 Å, which is less than the standard value of 4.177 Å, which also indicates the incorporation of Co^3+^ ions into the NiO crystal lattice.

Reduction of the obtained systems in the hydrogen flow proceeds according to reactions (1) and (2):NiO + H_2_ = Ni + H_2_O(1)
Co_3_O_4_ + 4 H_2_ = 3 Co + 4 H_2_O(2)

In the XRD patterns of the reduced Ni-enriched systems (Co1Ni2_W8 and Ni_W8), metallic cubic close-packed (ccp) phase reflexes are observed. In the Co1Ni2_W8 sample, the crystal lattice parameter (3.536 Å) is increased (Table 1) in comparison to the standard value of 3.524 Å for ccp Ni, which is associated with the incorporation of cobalt atoms in the Ni crystal lattice. As for Co-enriched samples, unusual XRD patterns are observed (Figure 1(IIa,b)). Diffraction peaks correspond to the hcp phase. However, analysis of the diffraction patterns for Co_W8 and Co2Ni_W8 samples shows that the 102_hcp_ peak is absent and the 101_hcp_ peak is abnormally broadened compared to the 100_hcp_ and 002_hcp_ peaks. For the Co2Ni1_W8 sample, 101_hcp_ broadening is even greater relative to the Co_W8 sample. For the Ni1Co1_W8 sample after reduction, reflexes of the ccp phase with a crystal lattice parameter of 3.532 Å are observed, while the 101_hcp_ reflex can also be detected in the XRD pattern.

Using the Co2Ni1_W8 sample’s composition as an example, we additionally investigated the effect of water as a co-solvent on phase transformations occurring during synthesis. Two additional samples of Co2Ni1_W0 and Co2Ni1_W4 were synthesized; a sample of Co2Ni1_W0 was obtained without using water and 4 vol.% of water was used for synthesis of the Co2Ni1_W4 sample. In the IR spectrum of the Co2Ni1_W0 sample (Figure 2a), peaks in acetate phase vibrations are observed [44]: 1026 cm^−1^, CH_3_ rocking; 1346 cm^−1^, CH_3_ symmetrical bending; 1407 cm^−1^, symmetrical CO_2_ stretching; 1556 cm^−1^, anti-symmetrical CO_2_ stretching. An increase in water content leads to a decrease in the intensity of acetate phase peaks (Figure 2b). In the IR spectrum of the Co2Ni1_W8 sample, there are practically no peaks in the acetate phases, and only peaks in the carbonate phase are observed at 1393 cm^−1^ (vibrations of various planar CO_3_^−^ ion modes) [45]. Thus, the addition of water as a co-solvent leads to the formation of carbonic acid, which effectively transforms acetate precursors into carbonates of the corresponding metals during SAS precipitation by reactions (3 and 4):Co(OAc)_2_ + H_2_CO_3_ = CoCO_3_ + 2 AcOH(3)
Ni(OAc)_2_ + H_2_CO_3_ = NiCO_3_ + 2 AcOH(4)

It was shown in [46] that the thermal decomposition of carbonate salts (Equation (5)) leads to the formation of more dispersed oxide phases compared to the decomposition of acetate salts (Equation (6)):6 CoCO_3_ + O_2_ = 2 Co_3_O_4_ + 6 CO_2_(5)
3 Co(OAc)_2_ + 8 O_2_ = Co_3_O_4_ + 12 CO_2_(6)

Figure 3 shows the XRD data of samples with the Co2Ni1 composition after calcination in air at 300 °C (I) and after reduction in a hydrogen flow at 300 °C (II); the samples were synthesized with different water content as a co-solvent. After calcination, only the spinel phase (Co_3_O_4_ or NiCo_2_O_4_) is observed for the Co2Ni1_W0 and Co2Ni1_W8 samples, whereas a cubic face-centered NiO phase is observed in addition to the spinel phase for the Co2Ni1_W4 sample. The structural characteristics of the samples are presented in Table 2. For the reduced Co2Ni1_W0 and Co2Ni1_W4 systems, reflexes of the defective hcp phase and the ccp phase are observed, whereas in the Co2Ni1_W8 sample, only reflexes of the defective hcp phase are observed. It is interesting to note that in the Co2Ni1_W0 and Co2Ni1_W8 samples, the oxide phase is represented only by the spinel phase; however, reduction of the Co2Ni1_W0 sample leads to phase separation into metallic phases with different symmetry, unlike the Co2Ni1_W8 sample. It is likely that the homogeneity of the metal phase after reduction is affected not only by the homogeneity of oxide composition after calcination, but also by the size of the oxide phase crystallites. Even though the Co2Ni1_W0 and Co2Ni1_W4 samples have different phase composition after calcination (Figure 3I), the structural characteristics of the reduced samples are almost identical (Figure 3II).

TEM images of the Co2Ni1_W8 and Co_W8 systems obtained after reduction demonstrate that both samples consist of spherical agglomerates not exceeding 150 nm in size, which form a branched morphology (Figure 4). Additionally, regions with a lack of crystal lattice ordering are observed in these samples, which agrees with the XRD data on the presence of defects in the Co2Ni1_W8, Co_W8, and Co2Ni1_W0 samples. The detected distances, 1.98 and 0.99 Å for the Co2Ni1_W8 sample, are assigned to interplanar distances d_101_ and d_202_ of the hcp structure. The 2.06 and 1.18 Å distances for the Co_W8 samples can be assigned to interplanar distances d_002_ and d_103_ of the hcp structure, but the 2.06 Å distance can be also assigned to d_111_ of the ccp structure. For the Co2Ni1_W0 sample, the 1.98 and 0.99 Å distances are assigned to interplanar distances d_101_ and d_202_ of the hcp structure.

In the Co2Ni1_W8 sample, in addition to interlinked agglomerates with a branched morphology that are characteristic for the Co2Ni1_W0 sample (see Figure 5a), isolated spherical agglomerates of metal particles with a diameter of about 200 nm are also observed (see Figure 5b). According to the elemental mapping data, Co and Ni atoms are uniformly distributed over the volume of the agglomerates, and these samples have no phase separation into Co- and Ni-enriched regions. This means that, despite the presence of both the hcp and ccp phases in the Co2Ni1_W0 sample, these phases have the same elemental composition, indicating the formation of two solid solutions. It should also be noted that the elemental composition of the samples, according to EDX data, is close to the theoretical loads at the synthesis stage (Appendix A).

### 3.2. Magnetic Characterization

Magnetic measurements were performed for the sample series after reduction. Characteristic M(H) hysteresis curves registered at room temperature (300 K) for the Co1Ni2_W8 and Co2Ni1_W8 samples are presented in Figure 6I. For all the samples under investigation, enclosure of the loops occurs at the maximum field of 15 kOe, which allows the coercivity of all samples to be analyzed (Figure 6II). Notably, in the field region above ~10 kOeM(H), dependencies tend to saturate. No additional analysis was needed to prove that saturation magnetization decreases with the substitution of Co to Ni, which is in good agreement with data on Co-Ni alloys [47]. Full M(H) hysteresis loops can be found in the Appendix A.

To calculate the magnetic anisotropy constant, the high-field magnetization dependencies *M*(*H*) were studied for fields >10 kOe using Akulov’s law for magnetization approaching saturation. According to [48], dispersed sample magnetization can be presented in the following way:(7)$$M\left(H\right)=Ms\left(1-D\frac{4K^{2}}{M_{S}^{2}H^{2}}\right).$$

Here, *K* is a magnetic anisotropy constant and *D* is a symmetry coefficient, which is equal to 1/15 for uniaxial anisotropy and 2/105 for cubic anisotropy. The high-field parts of the magnetization curves linearize quite well in the *M*:*H*^−2^ coordinates, proving the relevance and applicability of Aculov’s law (see Figure 6I right insert and Appendix A).

Calculation of the magnetic anisotropy constant for the Co_W8, Co_AC, and Co2Ni1_W8 samples was performed using a symmetry coefficient of D = 1/15, which is in agreement with XRD simulations for these systems (Table 3) as the major part of these samples is considered to have an hcp structure. For the Co1Ni2_W8 and Ni_W8 samples, the symmetry coefficient was taken to be D = 2/105 as they have a ccp structure. For the Co1Ni1_W8 sample, there is some discrepancy regarding the use of an hcp or ccp symmetry coefficient for magnetic anisotropy calculations. To take into account both the hcp and ccp phase contribution, the symmetry coefficient was taken to be D = (1/15 + 2/105)/2. Temperature dependencies of coercivity and remanent magnetization for all the samples can be found in the Appendix A.

Hysteresis loop parameters: coercivity, HC; remanent magnetization, MR; saturation magnetization, MS; and magnetic anisotropy constant, K. Dependencies in relation to Co/Ni ratio are presented in Figure 7a–d. Comparing the two pure cobalt samples, one can observe that the Co_W8 sample is magnetically harder since it has higher remanent magnetization than the Co_AC sample, which could be due to the higher defect ratio of the former sample. At the same time, a small addition of a magnetically softer element, Ni, does not immediately result in magnetically softer material. In fact, the highest coercivity among all the samples was registered for the Co2Ni1_W8 sample, both at 4.2 K and 300 K, while room temperature (300 K) remanent magnetization decreased insignificantly compared to the Co_W8 sample. Further increases in Ni content led to a decrease in coercivity and remanent magnetization for all samples. Saturation magnetization decreases almost linearly with decreasing Co content (Figure 7c), and the same applies to the dependence of the magnetic anisotropy constant on the Co/Ni ratio (Figure 7d). The value of the magnetic anisotropy constant for the Co1Ni1_W8 sample was calculated using the ‘mean’ value of the symmetry coefficient, D = (1/15 + 2/105)/2. In theory, the magnetic anisotropy value may lie within the 3.05 × 10^6^–5.8 × 10^6^ erg/cm^3^ range for pure fcc (D = 2/105) or ccp (D = 1/15) structures.

### 3.3. Simulation of the Imperfect Structure of Metallic Particles and Calculation of XRD Patterns

The XRD patterns of the Co_W8 and Co2Ni1_W8 samples exhibit reflections characteristic of a metallic Co⁰ phase with a hexagonal structure (PDF No. 00-005-0727 P63/mmc). However, a strong anisotropic broadening of the diffraction peaks is observed. The Bragg reflections, 101_hcp_ and 102_hcp_, are anomalously broadened compared to the 100_hcp_, 002_hcp_, and 110_hcp_ maxima. For comparison, Appendix A shows the experimental XRD pattern for the Co_W8 sample and the XRD pattern simulated for the Co⁰ crystallite with a perfect hcp structure. These features indicate the presence of stacking faults in the hexagonal structure, i.e., the appearance of cubic ABC fragments in the initial ABABAB sequence (hcp–ccp-type intergrowths) [10,49,50,51].

Agreement between the experimental and simulated XRD patterns was achieved for the statistical models with hcp–ccp-type intergrowths. The best fits achieved for the experimental XRD patterns of the Co_W8, Co2Ni1_W8, Co2Ni1_W0, Co1Ni1_W8, and Co_AC samples are shown in Figure 8. The fraction of the cubic stacking mode (W_2_) and the estimated thicknesses of fragments with a ccp and hcp structure are listed in Table 3. The generation of stacking faults results in broadening of the 101_hcp_ and 102_hcp_ maxima and does not affect the 100_hcp_, 002_hcp_, and 110_hcp_ peaks. An increase in the fraction of the cubic stacking mode intensifies broadening effects.

For the Co_W8 sample (Figure 4a), the relative amount of the cubic stacking mode in the hexagonal structure was determined to be about 35%. The estimated average thickness of ccp fragments is about 1 nm, which is consistent with HRTEM data showing thin stripes related to domains with different close-packed structures (Figure 4b). For the Co2Ni1_W8 sample (Figure 4b), a larger fraction of the cubic stacking mode was determined (43%). This was expected because the 101_hcp_ peak in the XRD pattern of the Co2Ni1_W8 sample was much broader. The average thickness of ccp fragments was the same, but the thickness of hcp fragments was decreased due to a higher fraction of the cubic sequence mode. These data suggest that doping the Co⁰ phase with Ni promotes the formation of a ccp structure. The ccp structure is intrinsic to the metallic Ni⁰ phase. These data agree with first-principles studies of Co-based binary alloys that revealed Ni favoring a cubic structure instead of a hexagonal one [52].

Experimental XRD patterns of the Co2Ni1_W0 and Co2Ni1_W4 samples are identical (Appendix A) but drastically differ from the XRD pattern of the Co2Ni1_W8 sample (Figure 1b). The 101_hcp_ peak broadens significantly. The peak at 51.5⁰, which relates to the 200_ccp_ reflection, becomes more intensive and narrower. This implies both an increase in the fraction of the cubic stacking mode and enlarged thickness. However, it was not possible to describe the experimental XRD pattern of the Co2Ni1_W0 sample by considering single-type particle models. A good fit was achieved only after considering an equimolar mixture of model particles of two types (Figure 8c, Table 3). The first type of particle is characterized by a higher fraction of the cubic stacking mode and the largest average thickness of cubic fragments (type I). The second type of particle contained thin cubic fragments at a lower concentration (type II). The model particle for the second type is close to that of the Co2Ni1_W8 sample. Metallic particles with differing domain thicknesses were found in HRTEM images of the Co2Ni1_W0 sample (Appendix A).

The simulation results for the Co1Ni1_W8 sample demonstrate that this sample is characterized by the one-type particle model, with a larger thickness of cubic fragments and a smaller thickness of hexagonal fragments (Figure 8d, Table 3).

It should be noted that the SAS precipitation process plays a major role in determining the structure of the obtained systems regardless of the subsequent stages of calcination and reduction. The Co_AC sample obtained by cobalt acetate calcination with a subsequent reduction also displays a broadening of the 101_hcp_ reflection in its XRD diffraction pattern (Figure 8e); however, this broadening is not as significant as in the Co2Ni_W8 and Co_W8 samples. The characteristics of the Co_AC sample obtained from the simulation are presented in Table 3. The size of the hcp phase domains for this sample is more than 3 nm, which is the largest value for all samples.

The obtained data revealed the impact of co-solvent water addition on the structure of formed Co2Ni1 nanoparticles. With high water content (8%), intergrowths in metallic Co2Ni1 particles are very thin (~1 nm), but the average thickness of hcp lamellar domains slightly exceeds the thickness of ccp lamellar domains. With low water content (4%) or no water addition, two types of particles are formed: one with thin lamellar domains (in the case of high water content) and one with wider ccp lamellar domains (the average thickness of ccp domains is almost twice of that of hcp domains). The heterogeneity of particles in terms of the concentration and distribution of defects increases with decreasing water content.

## 4. Conclusions

In this work, a method was proposed for the synthesis of mixed Co-Ni-containing bimetallic systems with a solid substitution solution structure based on the co-precipitation of acetate precursors in a supercritical carbon dioxide medium. It was shown that, in the case of Ni-enriched bimetallic systems, the formation of a cubic close-packed (ccp) structure occurs, whereas the formation of a defective hexagonal close-packed (hcp) structure occurs in Co-enriched systems. The Bragg reflections, 101_hcp_ and 102_hcp_, are anomalously broadened compared to the 100_hcp_, 002_hcp_, and 110_hcp_ reflections. This broadening was described using a model with a hexagonal close-packed structure packaged with embedded cubic fragments. An increase in Ni content leads to an increase in the fraction of cubic fragments in the system. It has also been shown that an increase in water content when used as a solvent leads to a decrease in the size of cubic fragments and a more uniform distribution of them. Study of the magnetic characteristics of the obtained systems showed that, for systems with a high amount of cubic fragments in the hcp structure, greater values for coercivity and remanence were observed.

## Figures and Tables

**Figure 1 nanomaterials-12-04366-f001:**
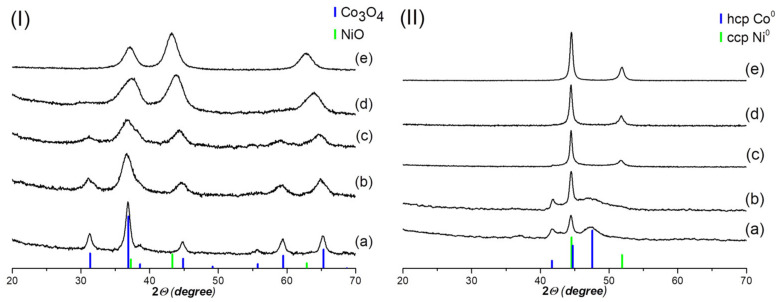
XRD patterns of obtained Co-Ni-containing systems after calcination (**I**) and reduction (**II**): Co_W8 (**a**); Co2Ni1_W8 (**b**); Co1Ni1_W8 (**c**); Co1Ni2_W8 (**d**); Ni_W8 (**e**).

**Figure 2 nanomaterials-12-04366-f002:**
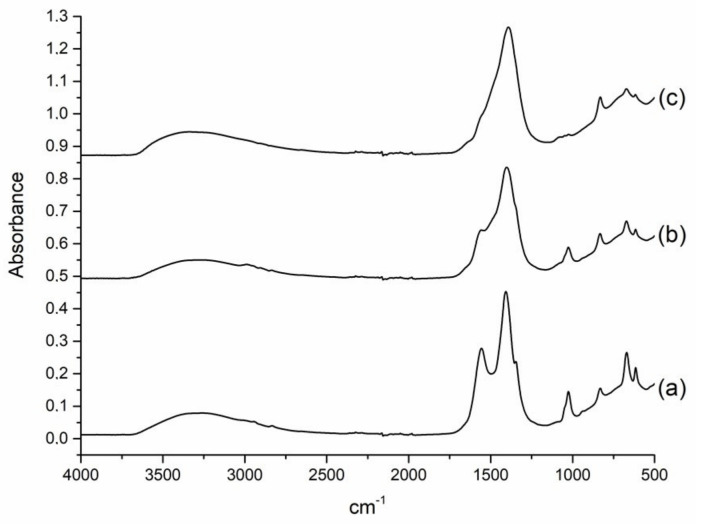
ATR FTIR spectra of the SAS-precipitated precursors of Co2Ni1 samples obtained with different water content: Co2Ni1_W0 (**a**); Co2Ni1_W4 (**b**); Co2Ni1_W8 (**c**).

**Figure 3 nanomaterials-12-04366-f003:**
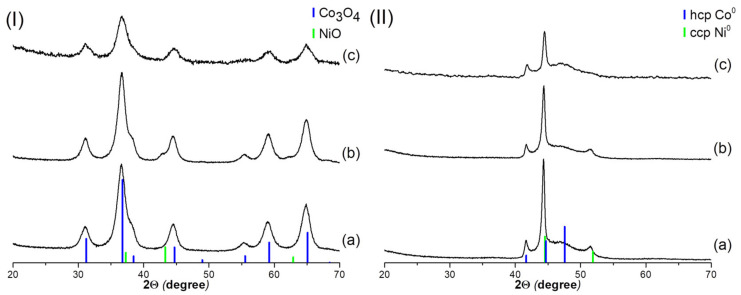
XRD patterns of Co2Ni1 samples obtained with different water content after calcination (**I**) and reduction (**II**): Co2Ni1_W0 (**a**); Co2Ni1_W4 (**b**); Co2Ni1_W8 (**c**).

**Figure 4 nanomaterials-12-04366-f004:**
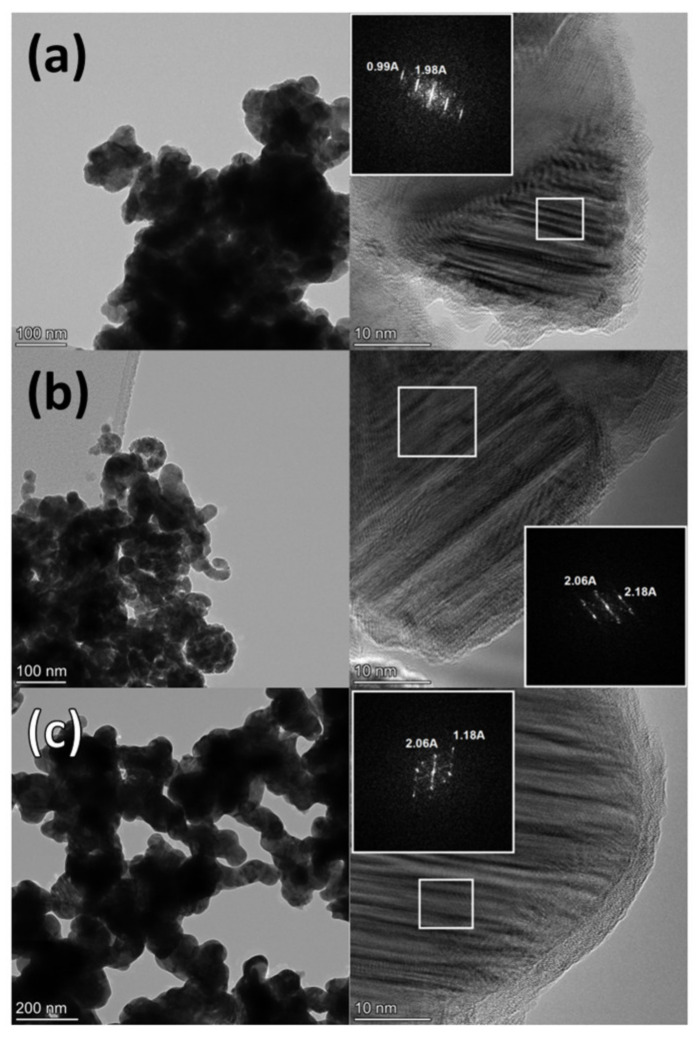
HR-TEM photographs and FFT patterns of the reduced samples: Co2Ni1_W8 (**a**); Co_W8 (**b**); Co2Ni1_W0 (**c**).

**Figure 5 nanomaterials-12-04366-f005:**
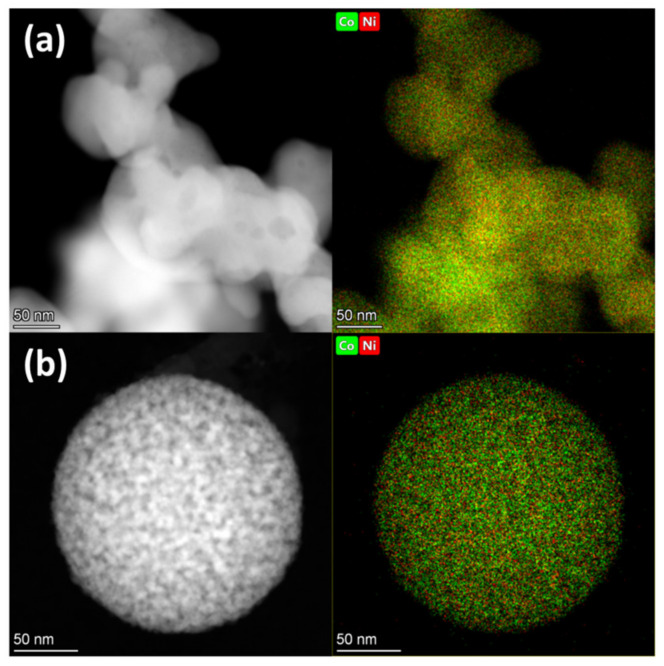
HAADF STEM and EDX elemental mapping of the reduced samples: Co2Ni1_W0 (**a**); Co2Ni1_W8 (**b**).

**Figure 6 nanomaterials-12-04366-f006:**
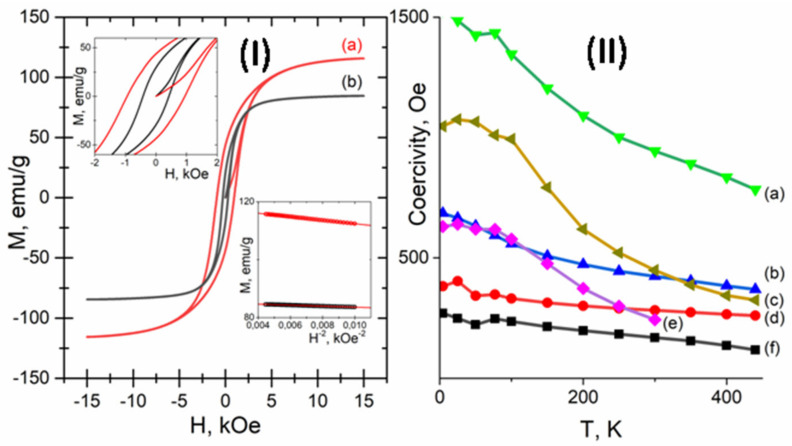
M(H) hysteresis loops at 300 K for the reduced samples (left insertion, M(H) near the origin; right insertion, high-field magnetization behavior in 1/H2 coordinates with linear regression): Co2Ni1 (**a**); Co1Ni2 (**b**) (**I**). Temperature dependencies of coercivity (HC(T)) for the samples: Co2Ni1_W8 (**a**); Co1Ni1_W8 (**b**); Co_W8 (**c**); Co1Ni2_W8 (**d**); Co_AC (**e**); Ni_W8 (**f**) (**II**).

**Figure 7 nanomaterials-12-04366-f007:**
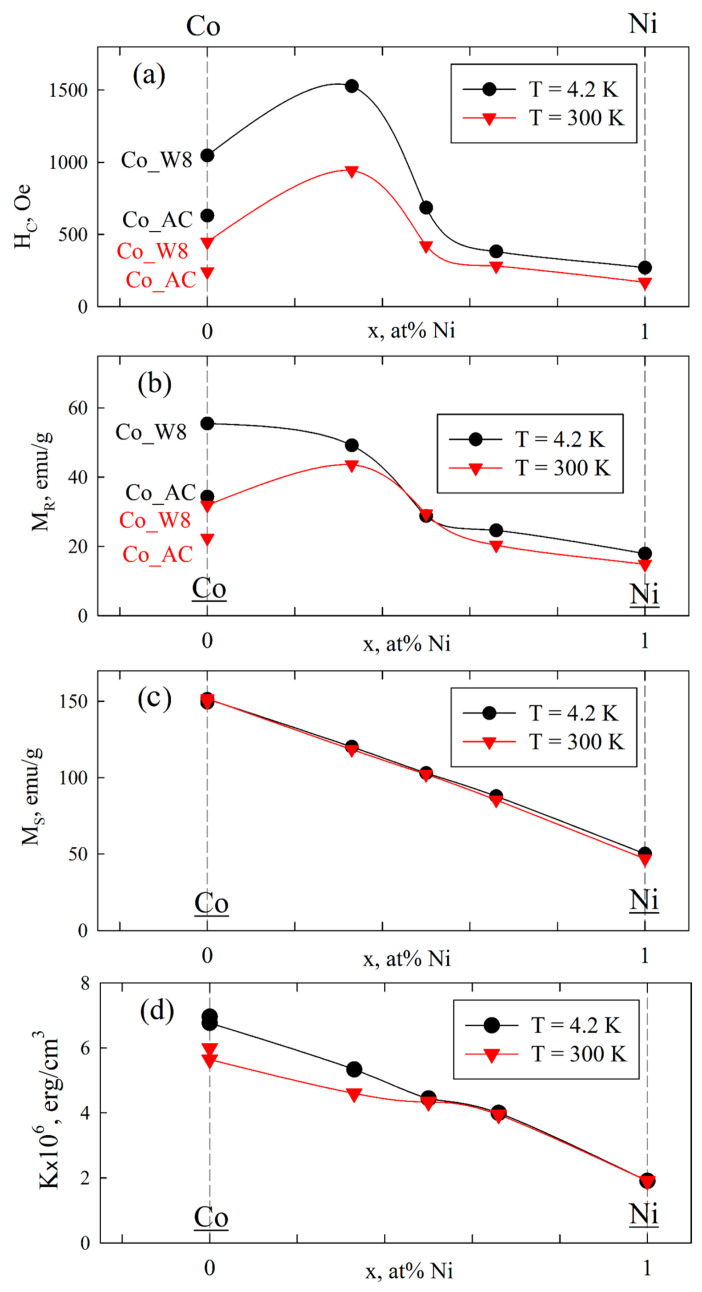
Concentration (x in Co_1−x_Ni_x_) dependencies for coercivity (HC) (**a**), remanent magnetization (M_R_) (**b**), saturation magnetization (M_S_) (**c**), and magnetic anisotropy constant (K) (**d**) for the studied series of samples at 4.2 and 300 K. Solid curves are visual guides. In (**a**,**b**), data for Co_AC and Co_W8 are marked in the graphs. In (**d**), data for Co_AC are shown without “guide lines”.

**Figure 8 nanomaterials-12-04366-f008:**
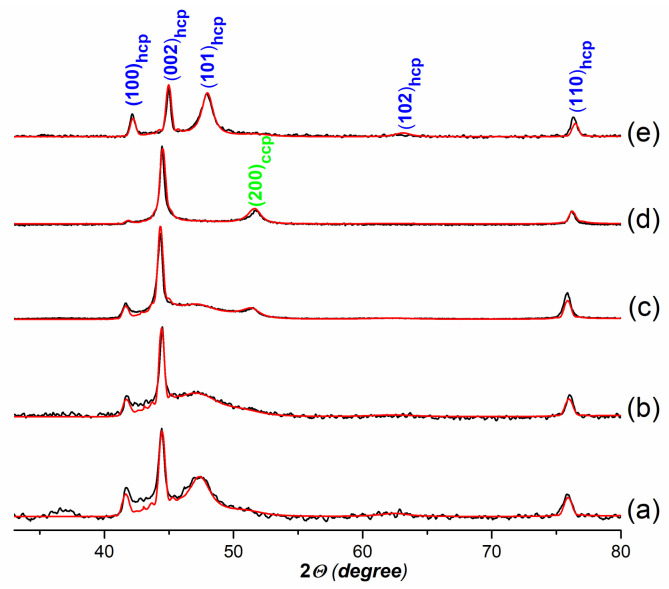
Experimental XRD patterns (black curves) and calculated XRD profiles (red curves) of the samples: Co_W8 (**a**); Co2Ni1_W8 (**b**); Co2Ni1_W0 (**c**); Co1Ni1_W8 (**d**); Co_AC (**e**).

**Table 1 nanomaterials-12-04366-t001:** Phase composition, lattice parameters, and DXRD size of Co-Ni-containing samples after calcination and reduction.

Sample		Phase	Lattice Parameter, Å	D_XRD_, nm		Phase	Lattice Parameter, Å	D_XRD_, nm
Co_W8	**Oxide Phases**	Co_3_O_4_	8.088	13.0	**Metal Phases**	hcp (defective structure)	a = b = 2.505, c = 4.070	25.0 (L_002_)
Co2Ni1_W8		Co_3_O_4_ or NiCo_2_O_4_	8.121	5.5		hcp (defective structure)	a = b = 2.500, c = 4.070	24.0 (L_002_)
Co1Ni1_W8		NiO	4.108	4.0		ccp	3.532	-
		Co_3_O_4_ or NiCo_2_O_4_	8.130	4.5		hcp (defective structure)	a = b = 2.498, c = 4.071	29.0 (L_002_)
Co1Ni2_W8		NiO	4.129	4.0		ccp	3.536	24.0
Ni_W8		NiO	4.177	4.5		ccp	3.524	22.5

**Table 2 nanomaterials-12-04366-t002:** Phase composition, lattice parameters, and DXRD size of Co-Ni-containing samples after calcination and reduction.

Sample		Phase	Lattice Parameter, Å	D_XRD_, nm		Phase	Lattice Parameter, Å	D_XRD_, nm
Co2Ni1_W0	**Oxide Phases**	Co_3_O_4_ or NiCo_2_O_4_	8.115	6.0	**Metal Phases**	hcp (defective structure)	a = b = 2.505, c = 4.060	20.0 (L_002_)
						ccp	3.544	20.0
Co2Ni1_W4		Co_3_O_4_ or NiCo_2_O_4_	8.112	6.5		hcp (defective structure)	a = b = 2.503, c = 4.060	20.0 (L_002_)
		NiO	4.185	6.0		ccp	3.544	20.0
Co2Ni1_W8		Co_3_O_4_ or NiCo_2_O_4_	8.121	5.5		hcp (defective structure)	a = b = 2.500, c = 4.070	24.0 (L_002_)

**Table 3 nanomaterials-12-04366-t003:** The fraction of the cubic stacking mode (W2) and the estimated average thickness of fragments with a ccp and hcp structure.

Sample	Fraction of Cubic Stacking Mode, W_2_ (%)	Estimated Average Thicknesses of Fragments	
		L_ccp_ (nm)	L_hcp_ (nm)
Co_W8	35	0.90	1.69
Co2Ni1_W8	43	0.90	1.19
Co2Ni1_W0	65 (type I)	2.03	1.09
	45 (type II)	0.90	1.09
Co1Ni1_W8	80	3.10	0.80
Co_AC	27	1.16	3.12

## Data Availability

Not applicable.

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
