# Peer review of "Synthesis of Co-Ni Alloy Particles with the Structure of a Solid Substitution Solution by Precipitation in a Supercritical Carbon Dioxide"

_nanomaterials, 2022, doi:10.3390/nano12244366_

Round 1

Reviewer 1 Report

Comments to the Author

The manuscript (nanomaterials-2035174) titled ‘Synthesis of Co-Ni alloy particles with the structure of a solid substitution solution by precipitation in a supercritical carbon dioxide’ has been carefully reviewed. The manuscript is a good contribution to investigate the structure and magnetic properties of bimetallic Co-Ni systems with different ratios of metals, obtained by the SAS method. The work was seriously discussed the effect of water on the phase composition of Co-Ni bimetallic systems. This context of article is suitable to the scope of this journal. The work is comprehensive and the data is analyzed and presented. Overall, the study is deserved to be published in this journal, but minor revisions needed to be done as follows.

1. As shown in Table 1, phase composition has been provided. However, the compositions of Co2Ni1_W8 and Co1Ni1_W8 samples were uncertain. Please determine the composition using inductively coupled plasma-optical emission spectrometer (ICP-OES).

2. In Figure 1, the hcp structure and ccp structure appeared in the reduced samples. What substances do they belong to? Metal, metal oxide or alloy?

3. In order to more clearly express the exposed crystal plane results, please mark interplanar spacing in TEM images, which was shown in Figure 4. The authors are encouraged to make further discussion on the TEM results.

4. As mentioned in the manuscript, the addition of water has an important impact on the composition of the Co-Ni alloy samples in this work. So please explain the main role of water in the synthesis process of the samples. In addition, please write the possible reaction equations involved in the sample synthesis under two different calcination conditions of calcination and reduction.

5. The languages should be carefully checked and polished.

Author Response

  1. As shown in Table 1, phase composition has been provided. However, the compositions of Co2Ni1_W8 and Co1Ni1_W8 samples were uncertain. Please determine the composition using inductively coupled plasma-optical emission spectrometer (ICP-OES).

Indeed, it was not possible to determine the phase composition in the samples of Co2Ni1_W8 and Co1Ni1_W8, since the phases of Co3O4 and NiCo2O4 are poorly distinguishable due to high dispersion. However, using the ICP-OES method will provide information about the elemental composition. It is not possible to obtain data on the phase state of the oxides using ICP-OES. To obtain data on the elemental composition of our samples, we used EDX analysis. This method has shown that the elemental composition of the samples is close to the theoretical loads at the synthesis stage. We have added EDX data to the supplemental materials (Figure S2).

  1. In Figure 1, the hcp structure and ccp structure appeared in the reduced samples. What substances do they belong to? Metal, metal oxide or alloy?

Thank you for this comment. The figure shows the positions of the XRD reflexes for the hcp-Co0 structure and ccp-Ni0 structure. We have added caption data to the Figure 1 and Figure 3.

  1. In order to more clearly express the exposed crystal plane results, please mark interplanar spacing in TEM images, which was shown in Figure 4. The authors are encouraged to make further discussion on the TEM results.

We have added FFT patterns, as well as supplemented the discussions of the results.

  1. As mentioned in the manuscript, the addition of water has an important impact on the composition of the Co-Ni alloy samples in this work. So please explain the main role of water in the synthesis process of the samples. In addition, please write the possible reaction equations involved in the sample synthesis under two different calcination conditions of calcination and reduction.

Thank you for this comment. In the text we have described the possible mechanism of the impact of water, as well as the equations of the transformation reaction at the stages of synthesis. We also cited a paper in which we observed similar effects on a monophase Co-containing system.

  1. The languages should be carefully checked and polished.

We tried to correct the language in our work.

1 Nesterov, N. S., Pakharukova, V. P., & Martyanov, O. N. (2017). Water as a cosolvent – Effective tool to avoid phase separation in bimetallic Ni-Cu catalysts obtained via supercritical antisolvent approach. The Journal of Supercritical Fluids, 130, 133–139. https://doi.org/10.1016/j.supflu.2017.08.002

2 Nesterov, N. S., Smirnov, A. A., Pakharukova, V. P., Yakovlev, V. A., & Martyanov, O. N. (2021). Advanced green approaches for the synthesis of NiCu-containing catalysts for the hydrodeoxygenation of anisole. Catalysis Today, 379, 262–271. https://doi.org/10.1016/j.cattod.2020.09.006

3 Marin, R. P., Kondrat, S. A., Pinnell, R. K., Davies, T. E., Golunski, S., Bartley, J. K., Hutchings, G. J., & Taylor, S. H. (2013). Green preparation of transition metal oxide catalysts using supercritical CO2 anti-solvent precipitation for the total oxidation of propane. Applied Catalysis B: Environmental, 140–141, 671–679. https://doi.org/10.1016/j.apcatb.2013.04.076

Reviewer 2 Report

The authors describe the synthesis of Co-Ni bimetallic particles using a supercritical SAS  technique. They obtained two types, Co-enriched and  Ni-enriched  particles. Co-enriched had hcp structure and  Ni-enriched  had ccp structure. The metal content depended on the amount of water in the preparation method.

Generally it is a clear piece of research, however, there are several issues to clarify:

Synthesis: an scheme of the equipment should be shown (at least in the S.I), that is the volume of the precipitation chamber?

 There is no mention of the solubility of the reactants in the solvents used, individually or in the mixture.

It is not clear the Co:Ni ratio before processing in SAS.

What is the effect on the oxidation at 300 C in the bimetallic samples, no sintering?

Samples were then reduced with H2 also at 300C. this is a harsh condition; does this have any advantage compared to other reducing methods such as hydrazine?

Is there any advantage of not using this material free instead of being supported?

Line 204: 1393 cm-1 carbonate phase, does this carbonation takes place during SAS?, that is, is acetate converted into carbonate with CO2 and water. Are there any mechanistic explanation for this?

Author Response

  1. Synthesis: an scheme of the equipment should be shown (at least in the S.I), that is the volume of the precipitation chamber?

Thank you for this comment. We have added installation descriptions and also added a scheme to the Supplemental information.

  1. There is no mention of the solubility of the reactants in the solvents used, individually or in the mixture.

We did not set out to determine the solubility of acetates, but we have chosen small concentrations to be sure of the solution stability. We have added comments on the selected concentrations in the synthesis description.

  1. It is not clear the Co:Ni ratio before processing in SAS.

Thank you for this comment. For a better understanding, we have added a description of the ratio of elements in the description of the system synthesis method.

  1. What is the effect on the oxidation at 300 C in the bimetallic samples, no sintering?

Control over sintering processes is very important in the synthesis of dispersed systems. We did not observe significant sintering of the samples at the calcination stage. In our case, sintering was observed at the reduction stage. Thus, the sizes of metal crystallites after reduction are larger than the sizes of oxide crystallites. These data are in good agreement with both our past work [1, 2] and the work of other researchers [3].

  1. Samples were then reduced with H2 also at 300C. this is a harsh condition; does this have any advantage compared to other reducing methods such as hydrazine?

Yes, indeed, there are various methods of reducing oxides. In the framework of this project, we plan to synthesize supported metal system to increase the particle dispersion. In the future, we planning to investigate the catalytic activity of these systems. Hydrogen reduction is a traditional method of producing metal catalysts, so we had chosen it for our data to be comparable with open literature.

  1. Is there any advantage of not using this material free instead of being supported?

The metal-support interaction can lead to changes in the reduction parameters, as well as to changes in the polymorphic composition of metal particles, therefore, at the initial stage we decided to investigate the free systems in order to exclude the influence of this interaction. In the future, we plan to obtain supported catalysts and investigate their properties.

  1. Line 204: 1393 cm-1 carbonate phase, does this carbonation takes place during SAS?, that is, is acetate converted into carbonate with CO2 and water. Are there any mechanistic explanation for this?

Thank you for this comment. In the text we have described the possible mechanism of the impact of water, as well as the equations of the transformation reaction at the stages of synthesis. We also cited a paper in which we observed similar effects on a monophase Co-containing system.

1 Nesterov, N. S., Pakharukova, V. P., & Martyanov, O. N. (2017). Water as a cosolvent – Effective tool to avoid phase separation in bimetallic Ni-Cu catalysts obtained via supercritical antisolvent approach. The Journal of Supercritical Fluids, 130, 133–139. https://doi.org/10.1016/j.supflu.2017.08.002

2 Nesterov, N. S., Smirnov, A. A., Pakharukova, V. P., Yakovlev, V. A., & Martyanov, O. N. (2021). Advanced green approaches for the synthesis of NiCu-containing catalysts for the hydrodeoxygenation of anisole. Catalysis Today, 379, 262–271. https://doi.org/10.1016/j.cattod.2020.09.006

3 Marin, R. P., Kondrat, S. A., Pinnell, R. K., Davies, T. E., Golunski, S., Bartley, J. K., Hutchings, G. J., & Taylor, S. H. (2013). Green preparation of transition metal oxide catalysts using supercritical CO2 anti-solvent precipitation for the total oxidation of propane. Applied Catalysis B: Environmental, 140–141, 671–679. https://doi.org/10.1016/j.apcatb.2013.04.076

Reviewer 3 Report

1.      The title of this paper is Synthesis of Co-Ni alloy particles with the structure of a solid substitution solution by precipitation in a supercritical carbon dioxide, but supercritical carbon dioxide is not shown in the abstract, please unify supercritical carbon dioxide, supercritical antisolvent precipitation and supercritical fluids (Page 1).

2.      Supercritical fluids in the keywords is not shown in the abstract (Page 1).

3.      The result of the magnetic characteristics is not explained in the abstract (Page 1).

4.      Page 4, lines 156-157: Why do you choose the stage of the oxide state (after calcination) and the metallic state (after reduction) for research? Is there any research value in other stages?

5.      Unify the format of the two figures in Figure 6 (Page 8).

6.      the image name is “…… after calcination - (I)”, “Co-W8-(a)” in Figure 1-5, but it changed to “(II) - Temperature ……”, “(a) - Co2Ni1-W8” in Figure 6, unify the format of the image name.

7.      Page 4, lines 171-172: Check the correspondence of form and text, the lattice parameter of NiO in the Co1Ni1-W8 sample is 4.129 Å, but it is 4.108 Å in Table 1.

8.      References are out of date, please supplement the literatures of the last three years.

Author Response

Review Report3

  1. The title of this paper is Synthesis of Co-Ni alloy particles with the structure of a solid substitution solution by precipitation in a supercritical carbon dioxide, but supercritical carbon dioxide is not shown in the abstract, please unify supercritical carbon dioxide, supercritical antisolvent precipitation and supercritical fluids (Page 1).

We added information about supercritical CO2 in the abstract.

  1. Supercritical fluids in the keywords is not shown in the abstract (Page 1).

We described that CO2 is in a supercritical state and added this to the abstract.

  1. The result of the magnetic characteristics is not explained in the abstract (Page 1).

We added results of the magnetic characteristics in the abstract.

  1. Page 4, lines 156-157: Why do you choose the stage of the oxide state (after calcination) and the metallic state (after reduction) for research? Is there any research value in other stages?

Yes, indeed, other stages of synthesis are also important for understanding the synthesis process as a whole. One of the goals for further research is to obtain bimetallic catalysts with the structure of a solid substitution solution without phase separation. The structure and properties of the metallic state depend on the characteristics of the oxide phases that are being reduced. Unfortunately, the substances after SAS precipitation have an amorphous structure [3], so we studied them only by IR spectroscopy.

  1. Unify the format of the two figures in Figure

6 (Page 8).6. the image name is “…… after calcination - (I)”, “Co-W8-(a)” in Figure 1-5, but it changed to “(II) - Temperature ……”, “(a) - Co2Ni1-W8” in Figure 6, unify the format of the image name.

Thank you for these comments. Indeed, for a better perception, we unified the format of the Figure 6 description . We believe that correction of Figure 6 is enough to answer comments 5 and 6.

  1. Page 4, lines 171-172: Check the correspondence of form and text, the lattice parameter of NiO in the Co1Ni1-W8 sample is 4.129 Å, but it is 4.108 Å in Table 1.

Thank you for noticing this error. Yes, indeed, the wrong value of the lattice parameter was written in the text. We have changed this to match the value from Table 1.

  1. References are out of date, please supplement the literatures of the last three years.

The following references of the last three years works have been added to the text. [4], [17], [21] in the manuscript.

1 Nesterov, N. S., Pakharukova, V. P., & Martyanov, O. N. (2017). Water as a cosolvent – Effective tool to avoid phase separation in bimetallic Ni-Cu catalysts obtained via supercritical antisolvent approach. The Journal of Supercritical Fluids, 130, 133–139. https://doi.org/10.1016/j.supflu.2017.08.002

2 Nesterov, N. S., Smirnov, A. A., Pakharukova, V. P., Yakovlev, V. A., & Martyanov, O. N. (2021). Advanced green approaches for the synthesis of NiCu-containing catalysts for the hydrodeoxygenation of anisole. Catalysis Today, 379, 262–271. https://doi.org/10.1016/j.cattod.2020.09.006

3 Marin, R. P., Kondrat, S. A., Pinnell, R. K., Davies, T. E., Golunski, S., Bartley, J. K., Hutchings, G. J., & Taylor, S. H. (2013). Green preparation of transition metal oxide catalysts using supercritical CO2 anti-solvent precipitation for the total oxidation of propane. Applied Catalysis B: Environmental, 140–141, 671–679. https://doi.org/10.1016/j.apcatb.2013.04.076
